# Vibrational Spectroscopy in Urine Samples as a Medical Tool: Review and Overview on the Current State-of-the-Art

**DOI:** 10.3390/diagnostics13010027

**Published:** 2022-12-22

**Authors:** Francesco Vigo, Alessandra Tozzi, Muriel Disler, Alessia Gisi, Vasileios Kavvadias, Tilemachos Kavvadias

**Affiliations:** 1Department of Biomedicine, University of Basel, Hebelstrasse 20, CH-4031 Basel, Switzerland; 2Department of Gynecology and Obstetrics, University Hospital of Basel Petersgraben 4, CH-4031 Basel, Switzerland; 3Faculty of Medicine, University of Basel, Petersplatz 1, CH-4001 Basel, Switzerland; 4Independent Researcher, CH-4031 Basel, Switzerland

**Keywords:** spectroscopy, urine, infrared spectroscopy, Raman spectroscopy, FTIR, SERS

## Abstract

Although known since the first half of the twentieth century, the evolution of spectroscopic techniques has undergone a strong acceleration after the 2000s, driven by the successful development of new computer technologies suitable for analyzing the large amount of data obtained. Today’s applications are no longer limited to analytical chemistry, but are becoming useful instruments in the medical field. Their versatility, rapidity, the volume of information obtained, especially when applied to biological fluids that are easy to collect, such as urine, could provide a novel diagnostic tool with great potential in the early detection of different diseases. This review aims to summarize the existing literature regarding spectroscopy analyses of urine samples, providing insight into potential future applications.

## 1. Introduction

The achievement of an early diagnosis is essential for success in the treatment of any disease, including in the oncological field. As a matter of fact, in the absence of clinically detectable signs and symptoms or efficient screening programs for several diseases, new parameters need to be evaluated. New ideas and techniques are evolving to fill this gap, which would otherwise apparently be impassable with traditional methods. Liquid biopsy is one of the latest and most rapidly emerging diagnostic platforms in medicine. The term was introduced by Pantel and Alix-Panabières to identify circulating tumor cells (CTCs) in patients’ blood. This approach was firstly pursued as an early detection method, and its use was extended to the assessment of benign or malignant disease in blood, as well as in other body fluids such as urine, saliva, cerebrospinal fluid (CSF), or pleural effusion [1,2]. Among these, urine analysis has gained interest in recent years as it is easy to collect, non-invasive, and familiar to the patient. Since metabolites can be found in urine, there is the possibility of gaining indirect information about the metabolism of several organs as well as any inflammatory or neoplastic processes [3,4].

In the last two decades, there has been scientific interest in vibrational spectroscopy (VS), an analytical method potentially applicable to urine. While spectroscopy investigates the interaction between electromagnetic radiation (light) and matter, VS identifies the molecular structure of the analyzed sample through its vibrational characteristics interacting with a beam source. The revealed spectrum represents a faithful representation of the unique molecular characteristics investigated [5]. This rapid, non-destructive, minimally invasive, and relatively inexpensive technique has shown to be promising in detecting biomarkers in different biofluids such as urine [6]. Hence, with the emerging progress in technology and machine learning, the analysis and classification of multiple parameters derived from spectroscopic analysis of urine have been facilitated, and its potential exponentially elevated. This paper aims to present the state of the art regarding the spectroscopic analysis of urine samples and to provide insight into its potential medical applications, summarizing what has been achieved so far and looking to the future.

### Basics of Vibrational Spectroscopy

Vibrational spectroscopy (VS) identifies the molecular structure and composition of a sample by studying its vibrational characteristics. When a sample is exposed to a beam source, its molecular bindings’ vibrational status can give absorption (studied by infrared spectroscopy, IR) or scattering (studied by Raman spectroscopy, RS) processes.

If a molecule is exposed to an infrared beam that matches in frequency with the natural vibration of the molecule, absorption occurs. In IR spectrometry, to describe it in an extremely simplified way, infrared light passes through a sample and the intensity of the transmitted electromagnetic radiation is measured at each frequency by a detector, resulting in a characteristic spectrum.

RS, on the other hand, is based on the fact that when monochromatic light (light having same frequency, i.e., laser) hits a molecule, most of it is transmitted without change (elastic, Rayleigh scattering), so reaches the detector unaltered. In a small percentage, it is scattered with a different frequency, higher or lower (inelastic, strokes and anti-strokes), and so with different wavelengths. The difference between the energy of the incident photon, from the source, and the energy of the stroke or anti-stroke photon will correspond to the vibratory motion of the molecular bond, and so to its structure.

Due to specific physical principles, transitions which have large Raman intensities often have weak IR intensities and vice versa. This contrasting feature allows vibrational motions that might not be active in IR to be analysed using RS.

The development of these techniques has followed the development of laser systems and methods that allow amplification and better spectra capture (surface-enhanced Raman, resonance Raman, tip-enhanced Raman, polarized Raman, ATR-FTIR, Near IR, etc.), as well as the use of computers and statistical computational systems (Fourier transformation), which help obtain spectra quickly and make them easily interpretable [5,7,8].

To make it easier to understand the terminology used in evaluating the performance of an instrument/diagnostic technique, we have briefly summarised some key concepts and definitions in a handy “Refresh box”.



## 2. Search Strategy

We performed literature research in Google Scholar, ResearchGate, and PubMed papers from January 1947 to December 2020 using keyword algorithms to identify manuscripts related to spectroscopy in urine samples; we also reviewed literature citations, following PRISMA guidelines. The search string in PubMed was (title/abstract): Spectroscopy AND/OR Vibrational Spectroscopy AND/OR Infrared Spectroscopy AND/OR Raman Spectroscopy AND/OR Fourier Transform Infrared Spectroscopy AND/OR FTIR AND/OR ATR-FTIR AND/OR SERS AND Urine AND/OR Biofluids.

The inclusion criteria covered any article that explicitly described the spectroscopic methods applied to urine, whether human, animal, or artificial. Exclusion criteria included case reports, letters to editors, publications in non-peer-reviewed journals such as congress papers, and studies with abstract only, or those where the language used was not English. We manually screened each record, removing those ineligible by the previously defined criteria. As reported in Appendix A, 169 original articles have been identified, 17 of which were reviews, 61 reporting the use of spectroscopy and urine analysis to identify diseases or clinical conditions in vivo, and 13 applied in the human oncological field. We could also divide them into those using artificial urine (22), and of human (116) or animal origin (17). In total, 91 papers described the use of a spectroscopic method related to its sensitivity and specificity in correctly identifying the most varied substances in urine previously prepared, not exclusively in vivo.

## 3. Analysis of the Literature

### 3.1. From the First Attempts to Current Performance

The first described attempt, in 1947, shows how it was already possible to identify, in the considerable time of ca. 10 min, quantities of androsterone, a product of hormonal metabolism, in concentrations of 0.5%, using a primordial IR spectrograph [9]. With the evolution of the technique and available tools, timing has been reduced and detection limits increased; in 2013, using FTIR (Fourier-transform infrared spectroscopy) it was possible to identify concentrations of Ibuprofen up to 0.77 μgml^−1^ in a few seconds [10] and in 2018, Turzhitsky et al. used SERS (surface-enhanced Raman spectroscopy) to reach a limit of detection for Morphin in water of 5 pg mL^−1^ (100 ng mL^−1^ in urine) in less than 2 min, extraction procedure included [11]. 

### 3.2. Forensic Medicine Applications

As urine is one of the ways to eliminate exogenous substances, both legal and not, associated with the increasingly limited detection equipment, sports medicine initially took advantage of spectroscopy more than other areas of medicine. This is how the use, mainly with the SERS technique, has been applied for the identification of drugs such as opioids (e.g., morphine, tramadol) [12,13,14], amphetamine [15], methamphetamine, methylenedioxy-methamphetamine (MDMA) [16,17,18,19,20,21,22], cocaine [23], cannabinoids [24,25,26], ketamine [27], antipsychotics such as clozapine [28], benzodiazepines [29,30], and legal but potential doping substances such as erythropoietin [31] phenylethanolamine A [32], or ephedrine [33].

These studies showed that VS had similar, if not superior, capabilities to standard methods in identifying toxic and/or prohibited substances in urine. As a result, the medico-legal field turned its attention to it.

A good example of this interest and of the transition from the laboratory to clinical practice in forensic medicine is the use of the ATR–FTIR (attenuated total reflectance—Fourier-transform infrared) spectrograph by Takamura et al. in 2019, to identify the sex of donors from small portions of urine. Dried urine traces of 101 donors, 61 males and 40 females, were analyzed. The spectrum indicated slight differences between the two sexes in the spectroscopical regions of 1710−1690 cm^−1^, 1395−1380 cm^−1^, and 1070−1050 cm^−1^. A brief clarification is, at this point, appropriate: red radiation is that part of the magnetic spectrum that has wavelengths between those of visible radiation, 13,000 cm^−1^, and those of microwaves, >1 cm^−1^. The spectrum obtained from VS is represented as a diagram where in abscissa are the wavelengths, and in ordinate is the intensity of absorption reported in percentage. By visual comparison of the given spectra in this study, these differences were not higher than the standard deviations, so it was infeasible to detect the donors’ sex. Subsequently, a multivariate statistical model was developed using mathematical methods as partial least-squares discriminant analysis (PSD) associated with a genetic algorithm. As a result, the improved discrimination performance enabled the finding of previously undetectable differences, so it could be possible to reach a specificity, sensitivity, and total discrimination accuracy for donor-wise discrimination of 0.99 ± 0.01, 0.92 ± 0.02, and 0.97 ± 0.01, respectively [34].

Recognizing and differentiating biofluids in forensics is increasingly crucial and decisive in the development of an investigation. As reported by W. R. Premasir, SERS spectra of 24 h dried samples of human blood, vaginal fluid, semen, saliva, and urine could provide good quality and reproducible characteristic SERS signatures leading to rapid identification of body fluids (Figure 1) [35]. 

Identifying urine from other biofluids as vaginal fluid or sperm was also the question in the research of Gregório et al. featuring FTIR (Fourier-transform infrared spectroscopy) [36]. The authors showed how the IR region from 1480 to 1800 cm^−1^ was particularly suitable for detecting remains of bodily fluids on the layers of the three different pads tested. Although urine was easily identified because of urea’s molecular vibrations, discrimination between semen and vaginal fluid was limited because of having both similar bands due to proteins within the studied range.

### 3.3. Non-Oncological Applications

The ability to obtain valid results from both a qualitative and quantitative point of view has also pushed the potential of these techniques in the identification of substances that are omnipresent in urine and play a role in the evaluation of the physiological functioning of the renal system, sometimes demonstrating greater accuracy and precision over standard methods. This is the case of urea [37], creatinine [38,39,40], glucose [41,42], ketone [43], and salts such as sulphate, phosphate [44], and oxalate [45,46]. Also present in urine are amino acids such as tyrosine [47], detected by Yedongo Yo et al. in 2018, or others such as phenylalanine, which, in the case of diseases, can occur in altered concentrations. Phenylketonuria is an illness that, if not detected early, can lead to severe consequences for newborns. An alternative, quick, and effective method to measure it has been proposed by Murugesan et al.: mixing urine samples with a polyvinylpyrrolidone (PVP)-stabilized silver colloidal (AgC) solution. Once evaporated on a solid substrate with a rough surface, this technique could improve SERS sensitivity, yielding an estimated recovery higher than 97%, especially if ZnO powder, able to remove uric acid, the main interfering factor in the phenylalanine spectroscopic band, was added to the mixture [48].

Because of the rapidity in precisely identifying almost all dissolved compounds, VS may also play a role in toxicological investigation, for example, in the detection of acephate [49], a strong pesticide, or for the monitoring of therapy with antibiotics such as sulfamethoxazole [50], cefalosporine (cefazolin, cefoperazone, cefotaxime, ceftriaxone and cefuroxime) [51,52], moxifloxacin [53], painkillers such as paracetamol [54], ibuprofen [10], and antihypertensive drugs such as propranolol [55]. Identifying exogenous substances connected to particular lifestyles can help monitor patients’ compliance. For example, spectrometric identification of biomarkers, such as thiocyanate [56], or nicotine metabolites, such as cotinine and trans-3′hydroxycotinine (3HC) [57], showed the ability to identify smoking patients from non-smokers. Additionally, in these cases, the procedures (SERS) involved the addition of media enhancing the spectroscopic signal: in the first case, a recyclable substrate based on gold–silver nanoparticles and silica; in the second case, a gold colloidal solution.

In parallel to the potential spectral identification of individual new substances in the urine, spectra analysis has developed, mainly thanks to the evolution of mathematical and computational methods applied to capture differences, such as to allow the division into categories and then obtain diagnoses. In case of infectious diseases, the rapidity in the correct identification of the pathogen is directly linked to the correct therapeutic approach and, therefore, to clinical improvement [58]. Although great strides have been made in microbiology and virology, especially concerning the recognition of bacteria and fungi [59,60], the gold standard remains the culture, with consequent longer time for diagnosis and delayed therapy start [61]. Four publications, two of which from in 2013, have shown how, thanks to the use of support vector machines (SVM) and multivariate statistical analysis, pathogens such as Escherichia coli and Enterococcus faecalis can be identified without the use of cultures, with a considerable time advantage, reaching an accuracy of 92% [62,63]. Similar results were reported in a study featuring partial least square-discriminant analysis in the identification of not only E. coli and E. faecalis but also of K. pneumoniaea and S. saprophyticus, with a >95% accuracy and >99% specificity, in less than 1 h, including the filtration and centrifugation procedure [64].

Using FTIR, Steenbeke et al. confirmed the power of spectroscopy in the detection of proteins, lipids, white blood cells (typical of pyuria), and red blood cells (typical of hematuria), as well as in the identification of crystals in urine (Figure 2). Using principal component analysis (PCA) to analyze the obtained spectra, they differentiated between Gram-negative and Gram-positive species, while soft independent modelling of class analogy (SIMCA) revealed promising classification ratios between the different pathogens (Figure 3) [65]. The power of classification of these techniques is of particular interest in cases where distinguishing different patient groups without obvious biomarkers is needed or when invasive methods should be avoided. This is the case of kidney donors, who, in Jingmao’s study in 2017, were correctly classified (accuracy 96.50%) according to the probability of rejection (known by previous biopsy) using only PCA and linear discriminant analysis (LDA) of the spectra obtained from 30 patients’ urine samples [66]. A previous pilot study developed by Somorjaia in 2000 also involved kidney illness and proved, despite a small sample of patients (N = 67), the ability to distinguish between normal renal transplants and rejected allografts, using an optimal region selector method based on the IR spectra of urine [67]. In 2015 the power to find abnormal kidney function related to acute kidney transplant rejection was confirmed by using SERS in 58 patients, suggesting this method could have an earlier and more specific value than the clinically used biomarker, serum creatinine [68]. In 2017, a research group resulting from a collaboration between researchers from Taiwan and the UK identified that spectral features such as 1668 cm^−1^, 1033 cm^−1^, and 1545 cm^−1^ might be useful urinary biomarkers of acute inflammatory renal injury, renal fibrosis, and Glomerulonephritis (GN). The study started with an animal model and then proved its quality in human urine. The intensity of the urinary 1545 cm^−1^ spectral marker, attributable to the amide II band of the peptide bonds of urinary protein, was consistently elevated in patients with crescentic GN-ANCA-associated vasculitis (N 24) compared to the healthy volunteers (N 11). It was also evident that moderate-to-severe GN patients with a GFR (glomerular filtration rate) of <60 mL/min/1.73 m^2^, both in active disease or in remission, had a higher 1545 cm^−1^ band intensity than GN patients with GFR ≥ 60 mL/min/1.73 m^2^, showing that this spectrometric biomarker may be a good indicator of the severity of renal dysfunction [69].

An ever-present topic around spectroscopy in urine is recognizing and measuring specific physiological components in urine, such as urea, creatinine, and glucose, because variation in their concentrations has significant clinical consequences. Recently, Bispo et al. used a dispersive Raman spectrometer to recognize and measure these substances to correlate them with the risk of kidney failure in diabetic and hypertensive patients. Comparing the differences in the intensities of several peaks, they properly calculated the difference in the concentration of the urine biochemical components in each group. They reported a progressive decrease in creatinine and urea (*p* < 0.05) and an increase in glucose (*p* < 0.05) as a biomarker of disease progression. The principal component analysis and the discriminating model build-up showed an overall classification rate of 70% [70].

A good example of the use of spectrometric biomarkers applied to the evaluation of a disease and its evolution is the association made by Yang et al. in 2018 between the wavelength of 1509 cm^−1^ and the presence of coronary occlusion greater than 70%. Trying to correctly classify cardiovascular patients operated on with percutaneous coronary intervention, unoperated, and healthy through the SERS analysis of urine, this wavelength allowed a sensitivity and specificity of 90% and 78.9%, respectively. A possible explanation, whose reasoning links from the spectrometric results back to the biochemical counterpart, was identified in the platelet-derived growth factor-BB (PDGF-BB), a molecule known to be associated with ischemic heart disease. The subsequent spiked urine experiment of this PDGF-BB showed that this substance was precisely associated with this spectrum variation. Moreover, the measured SERS spectra of all urine samples from 87 patients with coronary heart disease were compared with the clinical data provided by the hospital, and they revealed that the appearance of 1509 cm^−1^ in SERS spectra was in good agreement with the results of coronary angiography technology where cardiovascular congestion was above 70% (Figure 4). In this subgroup classification, sensitivity and specificity were 87.9% and 87.0%, respectively [71].

### 3.4. Oncological Applications

Published data on spectral analysis of the urine of cancer patients are sparse. In our research, we found 13 publications wherein spectroscopy was applied to human cancer pathologies. We report below the seven with the most significant clinical relevance. Paraskevaidi et al. used FTIR spectroscopy to analyze urine samples after 6–8 h of fasting from 10 ovarian cancer and 10 endometrial cancer patients and compared the spectra with similarly collected urine samples from control patients undergoing hysterectomy for benign conditions. Using algorithms such as partial least squares discriminant analysis (PLS-DA), principal component analysis with support vector machines (PCA-SVM) (Figure 5), and genetic algorithm with linear discriminant analysis (GA-LDA), the authors reported high levels of accuracy for both endometrial (95% sensitivity, 100% specificity, 95% accuracy) and ovarian cancer (100% sensitivity, 96.3% specificity, 100% accuracy). The responsible discriminating spectral peaks were in the wavelengths of proteins and nucleic acids [72].

Gynecologic cancer was also the target of the analysis of Lin et al. in 2019, who analyzed the spectra of 43 breast cancer patients and compared them with 48 healthy volunteers and 50 gastric cancer patients. Using urine spectra and linear discriminative analysis, they could achieve a diagnostic specificity of 87.5%, 95.8%, and 95.7%, respectively [73]. Del Mistro et al., using Raman spectroscopy, compared urine samples from nine prostate cancer patients with nine healthy controls and also reported differences in the intensity of the spectra. Their algorithms using principal component analysis (PCA) and linear discriminant analysis (LDA) led to a sensitivity of 100%, a specificity of 89 %, and an overall diagnostic accuracy of 95 %. The authors also used urine after fasting, which was frozen within 4–5 h after collection and, in order to remove traces of proteins (e.g., hemoglobin) potentially interfering with SERS analysis, centrifuged at 14,000 g at 4 °C for 15 min before measurement [74]. Two published studies compared oral cancer patients with healthy controls, both using Raman spectroscopy and PCA/LDA algorithms. Elulamai et al. used first-voided morning urine samples, which were stored in a refrigerator at 4 °C for a maximum of 48 h after collection and were thawed to room temperature before measurements, and reported a sensitivity and specificity of 98.6% and 87.1%, respectively, with an overall accuracy of 93.7% in right classification of the presented urine [75]. Jaychandran et al. reported on 50 oral cancer patients, comparing them to 21 healthy controls. They used urine samples collected between 9 and 11 o’clock, which were then stored in an ice box. Measurements were performed using Raman spectroscopy and the PCA/LDA analysis showed an accuracy of 90.5% [6]. In 2016, Feng et al. analyzed Raman-enhanced spectra from 68 nasopharyngeal cancer patients, 55 esophageal cancer patients, and 52 healthy volunteers. They reported a diagnostic sensitivity of 95.5%, 90.9% and 98.1% and specificity 97.2%, 98.2% and 95.7%, respectively. They used a modified method for detecting nucleosides in urine with the help of Raman spectroscopy (RS), while the samples had been treated with affinity chromatography and supplemented with gold as a substrate for the spectroscopy analysis [76]. Similarly, Wang et al. used a modified technique enhancing RNA detection in urine for Raman spectroscopic detection. In a training cohort of 43 prostate cancer patients, they could demonstrate 93% specificity, 95.3% sensitivity, and 94.2% accuracy [77].

## 4. Discussion

The results of our search revealed a growing interest in the use of the physical properties of urine, as well as the potential substances dissolved in it, to recognize physiological and pathological processes. Compared to other biofluids, urine shows great advantages: its collection is completely non-invasive and low-cost, it does not require special medical or laboratory expertise, it has a quite stable and pure composition since it is derived from the renal filtration process, and it can be used simultaneously for various diagnostic purposes [78]. Conversely, the collection of blood, saliva, tears, and other biofluids, such as amniotic fluid, is more invasive, often requires special transportation and manipulation (e.g., blood clotting), and needs specialized staff and/or expensive instrumentation. This work shows the potential of the application of VS on urine samples for various diagnostic purposes, with the potential of a further improvement in accuracy if associated with the developing computational method. The increasing use of artificial intelligence in daily life (e.g., marketing) is the basis for the subsequent widespread use in clinical routine. Unfortunately, the need to collect millions of data to train algorithms that can provide credible results is currently rarely met due to the scarcity of extensive generalized studies, usually inhomogeneous in methods and techniques. As shown in this paper, published studies present considerable variability regarding every step of the process, from the collection, processing, and storage of the samples (for example, urine after fastening or random samples, freezing, centrifuging, etc.) to the chosen spectroscopy approach (e.g., SERS versus ATR-FTIR), the different machine learning algorithms used, and the statistical processing applied. These factors contribute to a significant inhomogeneity of the existing databases, which limits their interactions. The next step to overcome this limit could be to develop standard data analysis methods at every process step. Only accurate pre-processed data and appropriate use of machine learning and AI methods can lead to accurate analysis and classification of raw data obtained from the spectroscopy and spread the use of this analysis tool. Of note, advances in algorithm and computational power allow for gaining results that were not achievable some years ago. Mostly from 2000 onwards, the use of computationally intensive statistical models and algorithms such as PCA (principal component analysis), PLS (partial least squares), GA (genetic algorithm) or LDA (linear discriminant analysis) led to highly accurate measurements in various clinical studies including oncological diagnoses, treatment, and prognosis. The possibility of using artificial intelligence to classify patients according to their state of health or the state of response to a therapy or a surgical treatment through the analysis of urine characteristics represents an opportunity, although not entirely feasible in the present, too precious to be missed. We are now at the beginning of a new era wherein AI will play a major role in assisting physicians in clinical practice. The first, encouraging results highlighted in this review should suggest redoubling our effort to improve the accuracy of AI applications, especially if coupled with such an inexpensive, non-invasive, easy-to-use analytical technique as vibrational spectroscopy.

## 5. Conclusions

The repeatability of the results and the ability to obtain them rapidly and with high accuracy with low-cost equipment makes vibrational spectroscopy in urine samples a perfect candidate for promising future research in the medical field. However, the standardization of the method, the initiation of larger translational and clinical studies as well as further exploration of the machine learning methods are needed to achieve the desired high levels of reliability and consistency.

## Figures and Tables

**Figure 1 diagnostics-13-00027-f001:**
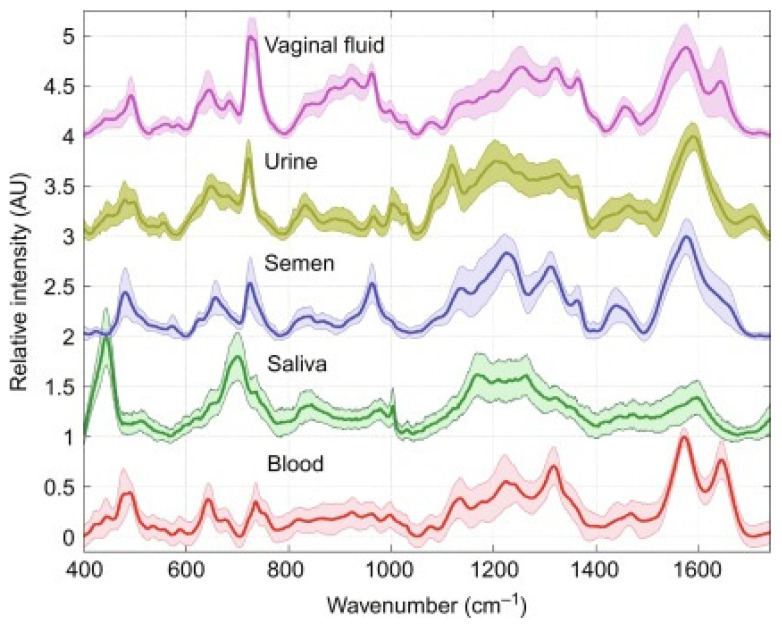
SERS spectra of dried human body fluids. Shaded regions correspond to standard deviation of 60 spectra (two donors of each body fluid type). Picture reproduced with permission from “*Frontiers and Advances in Molecular Spectroscopy*”, Jaan Laane, Elsevier, 2018 [36].

**Figure 2 diagnostics-13-00027-f002:**
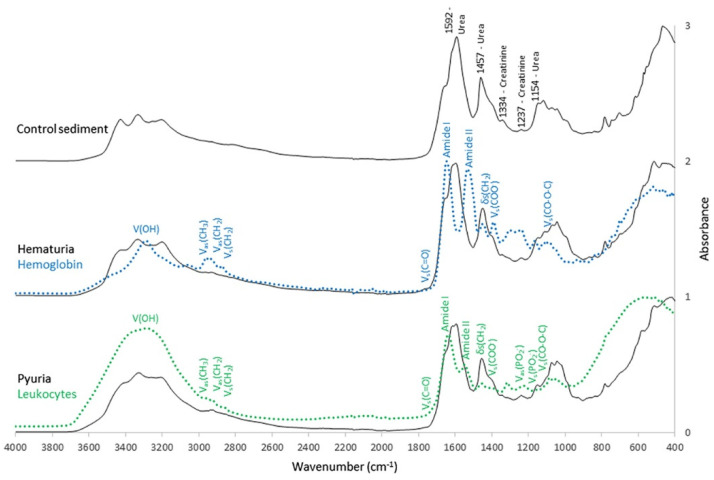
Mean normalized infrared (IR) spectra of urinary sediment of control subjects, patients with hematuria, and patients with pyuria. Distinctive peaks were present at 1592 cm^−1^ (urea), 1457 cm^−1^, (urea), 1334 cm^−1^ (creatinine), 1237 cm^−1^ (creatinine), and 1154 cm^−1^ (urea). The amide I (1655 cm^−1^ and amide II (1525 cm^−1)^ bands were overlapping in the region from 1510 to 1750 cm^−1^. A dried suspension of hemoglobin (blue line) and isolated leukocytes (green line) was added to the figure. Typical molecular assignments of a biological IR spectrum were indicated on the IR spectra of hemoglobin and the leukocytes: V = stretching vibrations; δ = bending vibrations; s = symmetrical vibrations and as = asymmetrical vibrations. Picture reproduced with permission from “*Exploring the possibilities of infrared spectroscopy for urine sediment examination and detection of pathogenic bacteria in urinary tract infections.*” Steenbeke, M. et al., Clinical Chemistry and Laboratory Medicine [65].

**Figure 3 diagnostics-13-00027-f003:**
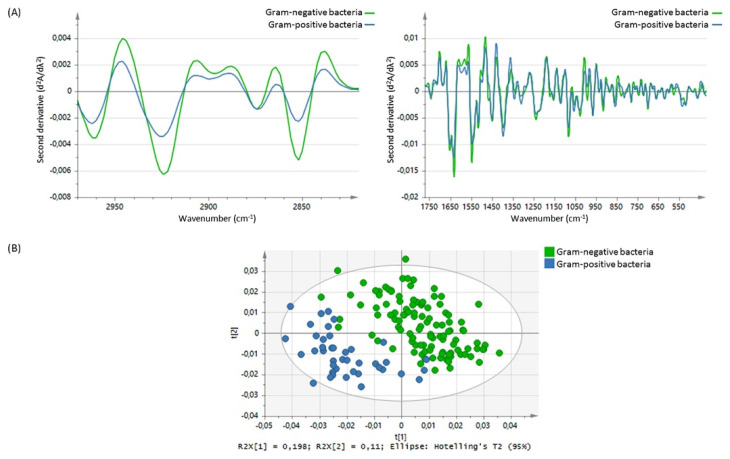
(**A**) Mean spectral differences in the second derivative from 2970 to 2820 cm^−1^ and from 1772 to 420 cm^−1^ between Gram-negative bacteria (green line) and Gram-positive bacteria (n = 41, blue line). (**B**) Score plot showing clustering of the Gram-negative bacteria (green dots) and Gram-positive bacteria (blue dots), based on principal component analysis in the 2970–2820 cm^−1^ and 1772–420 cm^−1^ spectral range. Picture reproduced with permission from “*Exploring the possibilities of infrared spectroscopy for urine sediment examination and detection of pathogenic bacteria in urinary tract infections.* Steenbeke, M. et al., Clinical Chemistry and Laboratory Medicine [65].

**Figure 4 diagnostics-13-00027-f004:**
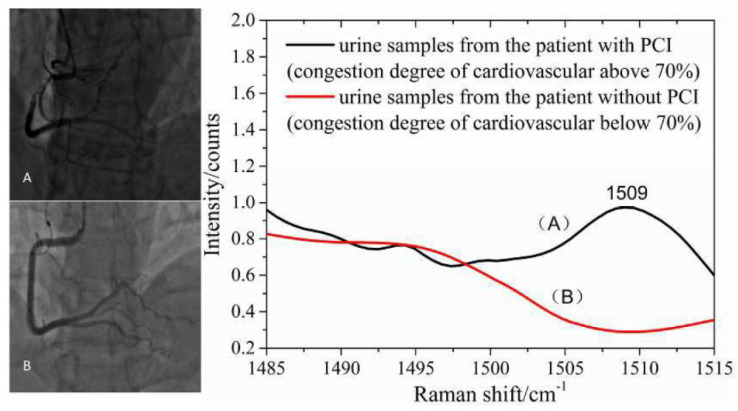
The SERS spectra of human urine could provide valuable information for noninvasive and prospective diagnosis of suspected cases of coronary heart disease. (**A**) urine samples from the patient with PCI (congestion degree of cardiovascular above 70%; (**B**) urine samples from the patient without PCI (congestion degree of cardiovascular below 70%). Picture reproduced with permission from “*Noninvasive and prospective diagnosis of coronary heart disease with urine using surface-enhanced Raman spectroscopy*”, Yang, H. et al., Analyst, 2018 [71].

**Figure 5 diagnostics-13-00027-f005:**
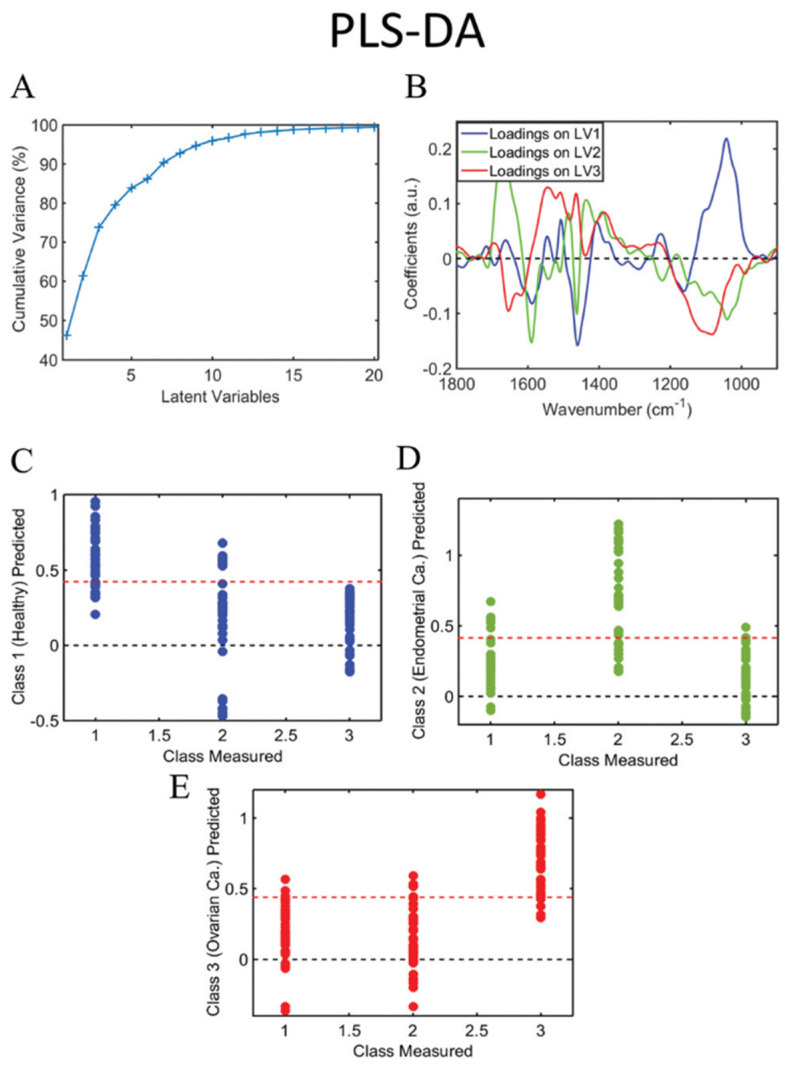
Cumulative explained variance using PCA (**A**); PCA loadings on PC1, 2 and 3 (**B**); predicted probability of healthy class versus endometrial and ovarian cancer (**C**); predicted probability of endometrial cancer class versus healthy and ovarian cancer (**D**); predicted probability of ovarian cancer class versus healthy and endometrial cancer (**E**). Class measured 1 = healthy control; 2 = endometrial cancer; 3 = ovarian cancer. PC: principal component. Picture reproduced with permission from “*Potential of mid-infrared spectroscopy as a noninvasive diagnostic test in urine for endometrial or ovarian cancer*”, Analyst, Paraskevaidi M. et al. 2018 [72].

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
