# Peer review of "Vibrational Spectroscopy in Urine Samples as a Medical Tool: Review and Overview on the Current State-of-the-Art"

_diagnostics, 2022, doi:10.3390/diagnostics13010027_

Round 1

Reviewer 1 Report

It is a nice review on urine analyses by molecular spectroscopic tools.  But it can be improved on its appealing to the prospective reader by decorating with some figures. I would like to see,

1. Add a reference on urine identification from other body fluid, 'Frontiers and Advances in Molecular Spectroscopy' by Jaan Lanne, Chapter 10.

2. Use figure 12 from above reference in section 3.2 to highlight the text.

3. Add a figure from Steenbeke et al or from Yang et al in 2018 or both work to section 3.3.

4. Add a figure from Paraskevaidi et al. work to section 3.4.

Reviewer 2 Report

The manuscript diagnostics-2067687 offers a thorough, yet concise, review of the existing literature regarding spectroscopic analyses of urine samples. 

The possibilities and limitations of the examined methods have been presented and discussed and I really appreciated the organization of Table 1. My comments here are mainly aimed at tightening up the clarity of what was communicated.

Since "a good figure is worth a thousand words", I wonder if the authors can prepare some original figures to convey the essential information of the review, summarising, for instance, the technological details or the research outcome.

As a review, the paper may be mainly read by inexperienced students or others starting their work in this specific field so every key concept should be better presented/highlighted. Some boxes, explaining for instance why these methods are called "vibrational spectroscopy", or the specific meaning of some terms like "accuracy" and "specificity" could be helpful.

In this sense, Tables 3 and 4 are not interesting/useful at all, and should be amended/removed: what is the meaning of "most frequently used" in table 3? Was it quantitative? If so, please report it. If not, please consider that there are a lot of differences between different techniques in the choice of preprocessing (even if the workflow can be similar, the preprocessing of FTIR spectra is often completely different from SERS spectra). As it is, I don't like the message delivered in Table 3. Please amend/expand it or remove it.

It is also not clear how table 4 was compiled. Many of the most used software/packages cited in Table 1 are missing (e.g. origin for spectroscopy, several R packages...).  Why? If the authors would arbitrarily select some of the available resources, they should declare it. Moreover, a link to the software webpage of every cited software/package should be added.

In conclusion, I recommend the publication in Diagnostics once the above-mentioned revisions are performed.
